# Growth-inhibiting effects of the unconventional plant APYRASE 7 of *Arabidopsis thaliana* influences the LRX/RALF/FER growth regulatory module

Shibu Gupta[1¤a], Amandine Guérin[1], Aline Herger[1], Xiaoyu Hou[1], Myriam Schaufelberger[1], Romain Roulard[2], Anouck Diet[1¤b¤c], Stefan Roffler[1], Valérie Lefebvre[2], Thomas Wicker[1], Jérôme Pelloux[2], Christoph Ringli[1] *

1 Department of Plant and Microbial Biology, Zurich-Basel Plant Science Center, University of Zurich, Zurich, Switzerland, 2 UMR INRAe BioEcoAgro, Biologie des Plantes et Innovation, Université de Picardie Jules Verne, UFR des Sciences, Amiens, France

¤a Current address: Securecell AG, Urdorf, Switzerland
¤b Current address: UFR Sciences du Vivant, Université Paris Cité, Paris Cedex 13, France
¤c Current address: Université de Reims Champagne-Ardenne, RIBP EA4707 USC INRAE 1488, SFR Condorcet FR CNRS 3417, Reims, France
* chringli@botinst.uzh.ch

**Data Availability Statement:** All relevant data are within the manuscript and its Supporting Information files.

## Abstract

Plant cell growth involves coordination of numerous processes and signaling cascades among the different cellular compartments to concomitantly enlarge the protoplast and the surrounding cell wall. The cell wall integrity-sensing process involves the extracellular LRX (LRR-Extensin) proteins that bind RALF (Rapid ALkalinization Factor) peptide hormones and, in vegetative tissues, interact with the transmembrane receptor kinase FERONIA (FER). This LRX/RALF/FER signaling module influences cell wall composition and regulates cell growth. The numerous proteins involved in or influenced by this module are beginning to be characterized. In a genetic screen, mutations in *Apyrase 7* (*APY7*) were identified to suppress growth defects observed in *lrx1* and *fer* mutants. *APY7* encodes a Golgi-localized NTP-diphosphohydrolase, but opposed to other apyrases of Arabidopsis, APY7 revealed to be a negative regulator of cell growth. APY7 modulates the growth-inhibiting effect of RALF1, influences the cell wall architecture and -composition, and alters the pH of the extracellular matrix, all of which affect cell growth. Together, this study reveals a function of APY7 in cell wall formation and cell growth that is connected to growth processes influenced by the LRX/RALF/FER signaling module.

## Author summary

The upright posture of land plants is made possible by solid walls that surround each individual cell. Hence, plant cell growth requires coordinated processes that expand the cell and, spatially and temporally controlled, the surrounding cell wall. On the molecular level, proteins involved in cell wall integrity (CWI) sensing monitor the status of the cell

**Funding:** This work was funded by grants by the Swiss National Science Foundation (Nrs. 31003A_166577 and 310030_192495 to C.R.). The funders had no role in study design, data collection and analysis, decision to publish, or preparation of the manuscript.

**Competing interests:** The authors have declared that no competing interests exist.

wall and influence and coordinate growth processes. Mutations in the genes encoding proteins of the CWI sensing machinery interfere with controlled cell growth, resulting in cells that are malformed, too small or big, or, as in root hairs, that even burst. In Arabidopsis, we found that the bursting of root hairs can be alleviated by changing a protein of the Golgi apparatus, APYRASE7. APYRASE7 appears to be a downstream target of the CWI sensing system since mutating the *APYRASE7* gene has an impact on the CWI sensing machinery, on cell growth, and the response to a peptide hormone regulating cell growth. Thus, our work provides a better insight into the different proteins involved in and necessary for controlled cell (wall) growth to take place.

## Introduction

The controlled expansion of plant cells requires a plethora of well-coordinated intra- and extracellular processes. These allow for spatially and temporally regulated enlargement of the cell wall surrounding plant cells, a process that requires cell wall material to be synthesized and integrated into the cell wall. Plants are equipped with a refined system to survey and modulate cell wall development by monitoring changes in cell wall composition and architecture. This sensing of cell wall integrity (CWI) involves a number of cell wall proteins and plasma membrane-spanning receptor kinases that help to perceive structural changes and convey this information to the cytoplasm [1,2]. Among these, the transmembrane *Catharanthus roseus* Receptor-Like Kinase1-Like proteins (CrRLK1Ls) have been well studied and identified as receptors of growth-regulating RALF (Rapid ALkalinization Factor) peptide hormones [3,4]. RALF peptides induce alkalinization of the apoplast, influence $Ca^{2+}$ dynamics, negatively influence cell growth, and modulate plant defense responses [5–8]. FERONIA (FER) is the best characterized member of the CrRLK1L family of Arabidopsis and is involved in numerous processes including pollen tube perception and rupture in the female gametophyte, plant immune responses, cell growth, and CWI signaling [7,9–12]. FER was identified as a receptor of RALF peptides in the context of plant immunity responses and control of cell elongation, where RALF-bound FER causes inactivation of the AHA2 proton pump and, hence, less acidic extracellular pH coinciding with reduced root growth [13,14]. In vegetative tissue, FER functions in a signaling process with Leucine-rich Repeat Extensins (LRXs) to regulate vacuole development, cell growth, salt stress responses, and defense activities [15–18].

LRXs are extracellular high-affinity binding sites of RALF peptides [19–21] tightly linked to the cell wall via their extensin domain that has the typical features of structural HRGPs (hydroxyproline-rich glycoproteins) [22]. Eleven *LRX* genes are encoded in the Arabidopsis genome, where *LRX8-11* are predominantly expressed in pollen, whereas the others are expressed in the different vegetative tissues. Interaction of most of the LRXs of vegetative tissue with FER has been demonstrated, suggesting an LRX/RALF/FER signaling module [15–18,21]. This is supported by the similar root hair phenotypes of a *fer-4* knock-out mutant and the *lrx1 lrx2* double mutant affected in the root hair-expressed *LRX1* and *LRX2*, respectively [17,23–25]. Also, the *fer-4* shoot phenotype is comparable to the shoot-expressed *lrx3 lrx4 lrx5* triple mutant [15,16,26]. While the role of LRXs in CWI sensing, cell wall formation, or salt stress is well established, much less is known about proteins that are involved in or modulated by this signaling activity downstream of the LRX/RALF/FER complex.

Cell growth processes are influenced by extracellular ATP (eATP) that is released by plant cells via Golgi-derived vesicles [27,28]. In Arabidopsis, eATP is bound by the plasma membrane-localized L-type lectin receptor kinase LecRK-I.9/DORN1 (DOes not Respond to

Nucleotides 1) and the closely related LecRK-I.5/P2K2, subsequently referred to as DORN1 and P2K2, respectively. Mutations in these genes affect eATP-induced $Ca^{2+}$ dynamics and the response to pathogens [29,30]. The amount of eATP is not only regulated by the vesicle-mediated export but also by Apyrases (AdenylPYRophosphatASES), nucleotide triphosphate phosphohydrolases that can hydrolyze ATP and other NTPs [27]. While many organisms, including plants, have ectoapyrases localizing to the cell wall, the seven members of the Arabidopsis Apyrase family localize to the endomembrane system of the Golgi or ER [31–33]. Downregulation of Golgi-localized APY1 and APY2 causes an increase in eATP, indicating that they possibly modify levels of Golgi-localized ATP destined for the extracellular space [34]. The APY1/APY2 downregulation also decreases cell expansion and root growth, interferes with pollen tube growth, alters the gene expression profile, and causes modifications of cell wall composition, hence has pleiotropic effects on plant development [34,35]. *apy6 apy7* double mutants have been shown to be affected in pollen grain formation and pollen tube growth including deformation of the exine cell wall [36].

Here, we identified APY7 as a modulator of the LRX/RALF/FER signaling module. Mutations in *APY7* alleviate defects in root hair development induced by mutations in *lrx1* and *lrx2* and alter defects developing in *fer* mutants. Opposed to other apyrases, APY7 is a negative regulator of cell growth and appears necessary for the growth-inhibiting effect of RALF1. APY7 abundance is regulated by Reactive Oxygen Species (ROS), has an impact on cell wall composition, and influences the sensitivity towards eATP. These findings reveal functions of APY7, a non-conventional Apyrase with a particular protein structure and atypical effects on cell-growth processes, compared to other APYs. They also demonstrate that the Golgi-localized APY7 has a significant impact on several aspects of cell growth processes that are connected to the LRX/RALF/FER signaling module.

## Results

### *rol16* suppresses the *lrx1* mutant root hair phenotype

A genetic approach was used to identify genes encoding proteins that play a role in the LRX/RALF/FER signaling module involved in CWI sensing and cell growth. To this end, a suppressor screen was performed on the *lrx1* mutant of *Arabidopsis thaliana* that is impaired in the root hair-expressed *LRX1* gene and develops short, branched, or burst root hairs [23] (Fig 1A). *lrx1* mutant seeds were treated with the mutagen ethyl methanesulfonate (EMS) and *rol* (*repressor of lrx1*) mutants were identified in the M2 generation based on a suppressed *lrx1* root hair phenotype. Seedlings of the *lrx1 rol16* mutant identified in this screen show re-establishment of root hair development and, thus, a wild type-like appearance (Fig 1A). Backcrossing of the *lrx1 rol16* mutant with *lrx1* resulted in the F2 generation in a 3: 1 segregation (42: 19 *lrx1*: wild type-like root hair formation), indicating that the *rol16* mutation is recessive.

### *rol16* is an allele of the ubiquitously expressed *APYRASE7*

To identify the *rol16* mutation, 15 F2 seedlings with wild type-like root hair development were isolated of the segregating population mentioned above, the material was pooled, and DNA was isolated for whole-genome sequencing (WGS). The WGS data of the *lrx1 rol16* mutant was compared to the already existing WGS data of *lrx1* [37] and an SNP with high coverage in the *lrx1 rol16* WGS dataset was identified in the *APYRASE7* (*APY7*, At4g19180) gene (S1A Fig). The SNP was confirmed by sequencing and a molecular marker for this SNP (S1B Fig) was created and tested on F2 seedlings of the backcross mentioned above. Over 50 F2 seedlings showing the *lrx1 rol16* phenotype were selected and revealed complete linkage of the mutation in *APY7* with the *rol16* mutant phenotype, suggesting that *rol16* is an *apy7* allele. The *rol16*

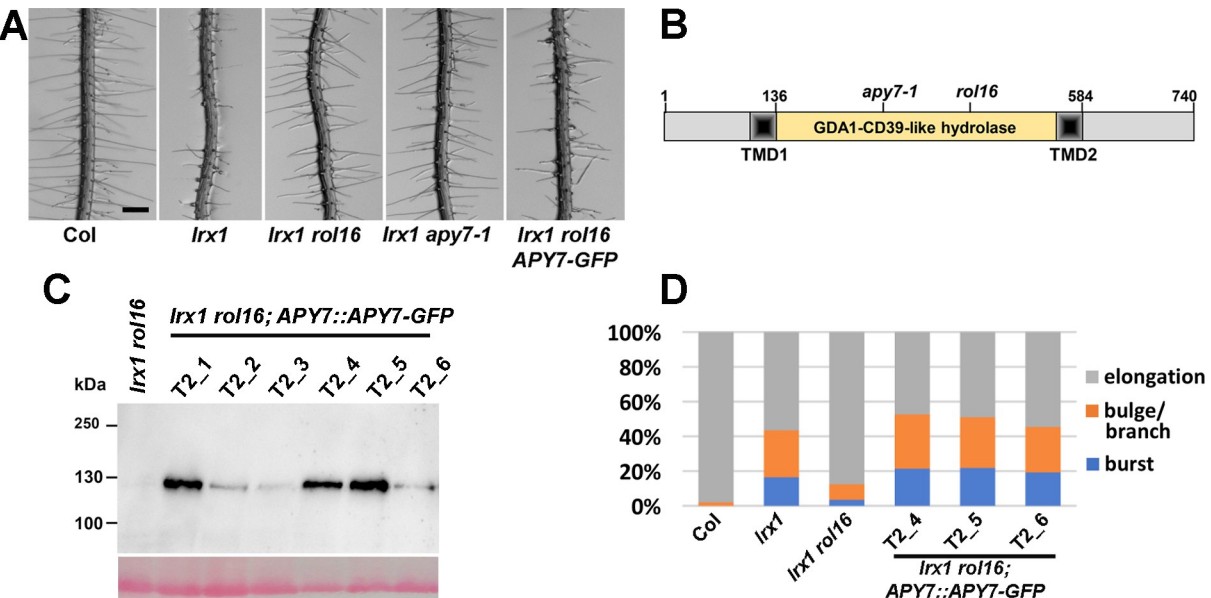

**Fig 1. *rol16* suppresses the *lrx1* root hair defect.** (A) Seedlings grown for six days in a vertical orientation show a defect in root hair development of the *lrx1* mutant compared to the wild type (Col). This defect is suppressed by *rol16* and by *apy7-1*, a second allele of *APY7*. *APY7::APY7-GFP* successfully complements the *rol16* mutation, resulting in an *lrx1*-like root hair phenotype. Bar = 0.5 mm. (B) *ROL16* codes for APY7 (Apyrase 7). Apyrases have a nucleoside phosphatase domain, referred to as GDA1-CD39-like domain. APY7 is one of two apyrases of Arabidopsis to have two transmembrane domains (TMD). Positions corresponding to *apy7-1* (SALK_124009) and *rol16* (Trp427stop) on the DNA level are also indicated. (C) Total protein extracts of seedlings expressing APY7-GFP were separated by SDS-PAGE and immunoblotted using an anti-GFP antibody. Lower panel shows PonceauS staining of the membrane. (D) Quantification of complementation of the *lrx1 rol16* root hair phenotype in three independent transgenic lines with different APY7-GFP levels. The three categories are elongating root hairs, enlarged/branched root hairs, and burst root hair initials, all in % of total root hairs of a given genotype (n>250).

allele contains a G to A mutation introducing a stop codon in APY7 at position Trp427 (Fig 1B). The *apy7-1* allele, containing a T-DNA insertion in *APY7* at the position corresponding to I263 (SALK_124009) [36], was crossed with *lrx1* and the subsequently identified *lrx1 apy7-1* double mutant also suppressed the *lrx1* root hair phenotype (Fig 1A), further supporting the assumption that *ROL16* is *APY7*. Quantitative RT-PCR on total RNA extracted from 10 days-old seedlings revealed a reduction but not depletion of the *apy7* mRNA in the *rol16* mutant compared to the wild type (S1C Fig). In view of the interruption of the coding sequence in the mutants, both alleles can be considered loss-of-function alleles despite remaining RNA levels.

APY7 has a GDA1-CD39 (nucleoside phosphatase) domain characteristic of apyrases [38] that is flanked by transmembrane domains (TMD). In Arabidopsis, APY7 and APY6 both have two TMDs, whereas all other APYs have only one TMD (Figs 1B and S1D). Five function-ally important sequence motifs conserved among apyrases of diverse origins are also present in APY7 (S1D Fig). Furthermore, APY7 has a C-terminal extension of around 110 amino acids not found in any other Arabidopsis apyrase, suggesting that APY7 might have a particular role among the Arabidopsis Apyrase proteins [27].

A complementation experiment was performed by transforming the *lrx1 rol16* double mutant with an *APY7::APY7-GFP* construct. Seedlings of the T2 generation of several indepen-dent transgenic lines developed *lrx1*-like root hair phenotypes (Fig 1A), demonstrating suc-cessful complementation of *rol16*. For detection of the recombinant APY7-GFP fusion protein, one hundred seedlings of each of these T2 families were grown for eight days on half-strength MS plates, pooled, and total protein was extracted. After separation of the proteins by SDS-PAGE, immunoblotting using an anti-GFP antibody revealed varying amounts of

APY7-GFP detected among the different T2 lines (Fig 1C). The effectiveness of complementation by different APY7-GFP levels was assessed by quantification of the root hair development in progenies of the transgenic lines T2-4, -5, -6, accumulating different amounts of APY7-GFP. The classification into root hairs that elongate, are enlarged or branched, or burst after formation of the initial bulge, revealed useful to describe and quantify the *lrx1* root hair phenotype. This revealed comparable complementation of different levels of APY7-GFP (Fig 1D). Together, these results demonstrate that *rol16* is indeed an allele of *APY7*.

The tissue specificity was analyzed in wild-type plants transformed with an *APY7::GUS* reporter construct and in the *APY7::APY7-GFP* transgenic lines. GFP fluorescence and the blue coloring induced by the GUS activity was observed at various stages of plant development and in different tissues including root hairs (S2 Fig), in line with the previous finding of *APY7* being expressed throughout the plant [36].

### *rol16* shows suppression of *lrx1 lrx2* and *fer* mutant phenotypes

Several LRXs of vegetative tissues, including LRX1, have been shown to function in a signaling process with FER, and the accumulation of mutations in *LRX* genes induces *fer*-like phenotypes [15–18]. As LRX1 and LRX2 function synergistically in root hair development, the *lrx1 lrx2* double mutant shows a severe root hair defect [24] comparable to the *fer-4* knock-out mutant [25]. To test whether *rol16* is able to suppress the *lrx1 lrx2* double mutant phenotype, an *lrx1 lrx2 rol16* triple mutant was established. Seedlings of this line indeed develop wild type-like root hairs (Fig 2A). Similarly, *rol16* largely suppresses the root hair defect of *fer-5*, a *fer* allele inducing an intermediate root hair phenotype [25] (Fig 2). In the virtually root hair-less *FER* knock-out allele *fer-4*, *rol16* allowed more root hairs to successfully enter the elongation phase (S3A Fig). Hence, *rol16* caused a partial suppression of the *fer-4* mutant root hair phenotype.

In the process of identifying a *fer-4 rol16* double mutant, a distorted segregation of the *fer-4* mutation was observed. Among the progeny of a wild-type plant heterozygous for *fer-4* ($ROL16^{+/+} fer-4^{+/-}$), 60 out of 800 seedlings showed a *fer-4* mutant phenotype. This less-than-25% frequency of $fer-4^{-/-}$ was expected considering the function of FER in the fertilization process, which reduces transmission of the *fer* mutation through the female gametophyte [39]. In a *rol16* mutant population segregating for *fer-4* ($rol16^{-/-} fer-4^{+/-}$), however, only 12 out of 800 seedlings were homozygous for *fer-4*. *rol16* does not affect fertilization efficacy, since in the progeny of a heterozygous *rol16* plant, almost one quarter revealed to be homozygous for *rol16* (S3B Fig). Hence, the observed strong reduction in the transmission of *fer-4* in the *rol16* mutant background must lay in a synergistic interaction between the two mutations. As a consequence of reduced fertility, *rol16 fer-4* mutants produce shorter siliques with less seeds than the respective single mutants (S3C Fig). In summary, the alleviation of the root hair defects in the *lrx1 lrx2*, *fer-4*, and *fer-5* mutants as well as the impact of *rol16* on fertility of the *fer-4* mutant suggest a genetic relationship between *FER* and *ROL16/APY7* and show that APY7 has an impact on processes that are influenced by LRX and FER proteins.

### APY7 is a negative regulator of cell growth required for RALF1-induced root growth inhibition

The LRX/RALF/FER signaling module modifies cell growth, which prompted us to test whether cell growth is altered in the *apy7* mutants. Indeed, *rol16* and *apy7-1* seedlings grew longer roots than the wild type and had longer epidermal cells (Fig 3A and 3B). Since LRXs and FER are RALF receptors [13,15,16,18,19], it was tested whether APY7 is involved in RALF1-mediated growth inhibition. When grown in the presence of 1 μM RALF1 peptide,

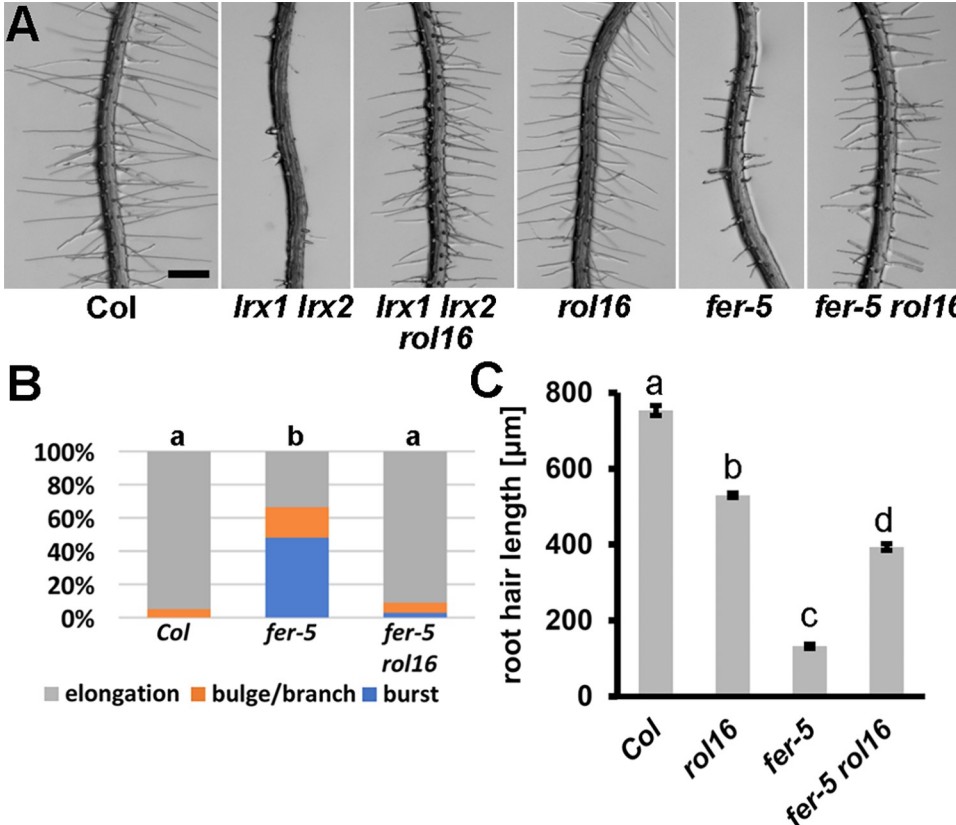

**Fig 2. *rol16* suppresses the *lrx1 lrx2* and *fer-5* root hair phenotype.** (A) Seedlings grown for six days in a vertical orientation show the strong root hair defect in the *lrx1 lrx2* double mutant that is fully suppressed by *rol16*. *fer-5* contains a T-DNA insertion after the kinase domain and develops an intermediate root hair defect which is suppressed by *rol16*. Bar = 0.5 mm. (B) Classification of different types of root hair defects in the wild type (Col) compared to *fer-5* and *fer-5 rol16*. Letters indicate significant differences in all three categories (n>250, student's t-test p<0.001). (C) Length of root hairs of the different lines. Letters above the graphs show statistically significant differences (n>90; student's t-test, p<0.001).

seedlings of the wild-type Col showed a reduction in root length, whereas seedlings of *fer-4*, *rol16*, or *apy7-1* mutants showed no inhibition of root growth (Fig 3C). This suggests that APY7 has a function in growth processes that are influenced by RALF1. APY7 negatively influences root growth and cell elongation and is required for RALF1-induced inhibition of cell growth processes.

## APY7 influences the apoplastic pH in root tissue

An important effect of RALF1 in regulating root growth is the alkalinization of the apoplastic pH, which inhibits root growth [5,13]. Since the *rol16* mutant showed reduced sensitivity towards RALF1, the pH was investigated in *rol16* and *apy7-1* using the ratiometric pH indicator dye HPTS [40]. In agreement with previous publications [40], an increase in the apoplastic pH was observed in Col seedlings upon RALF1 treatment (Fig 4A and 4B). Interestingly, even in the absence of RALF1, *rol16* and *apy7-1* mutant seedlings showed a more alkaline pH comparable to Col after RALF1 treatment (Fig 4A and 4B). Hence, inactivation of APY7 causes a more alkaline pH in the extracellular matrix. This finding urged us to investigate whether there is a correlation between increased pH in the cell wall and suppression of the *lrx1* root hair phenotype. To this end, we used a mutant affected in the proton pump *AHA2* important

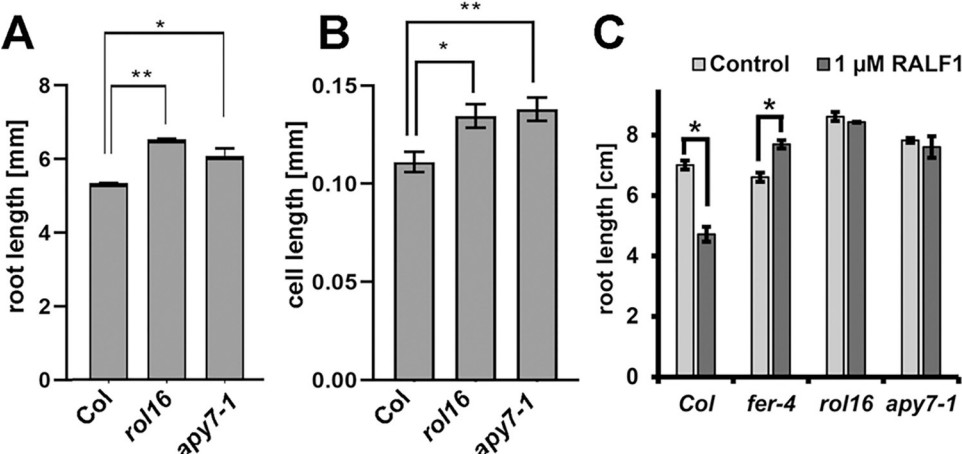

**Fig 3. *rol16* and *apy7-1* show increased cell growth and reduced response to RALF1.** (A) Primary roots length was measured in *rol16* and *apy7-1* mutant seedlings, which were significantly longer compared to the wild type (Col). (B) Cell length of trichoblasts was measured in the differentiation zone of the roots and revealed larger cells in the *rol16* and *apy7-1* mutants compared to the wild type (Col). (C) Seedlings were germinated on agar plates for three days and subsequently transferred to liquid medium containing 1 μM RALF1 peptide for three days. Wild-type Col seedlings responded to RALF1 by with reduced root growth. *rol16* and *apy7-1* mutants showed insensitivity towards RALF1. Asterisks indicate statistically significant differences in root length (student's t-test, n>10, p<0.01). Error bars represent standard deviation (n>5, student's t-test, p = * < 0.05, ** < 0.005).

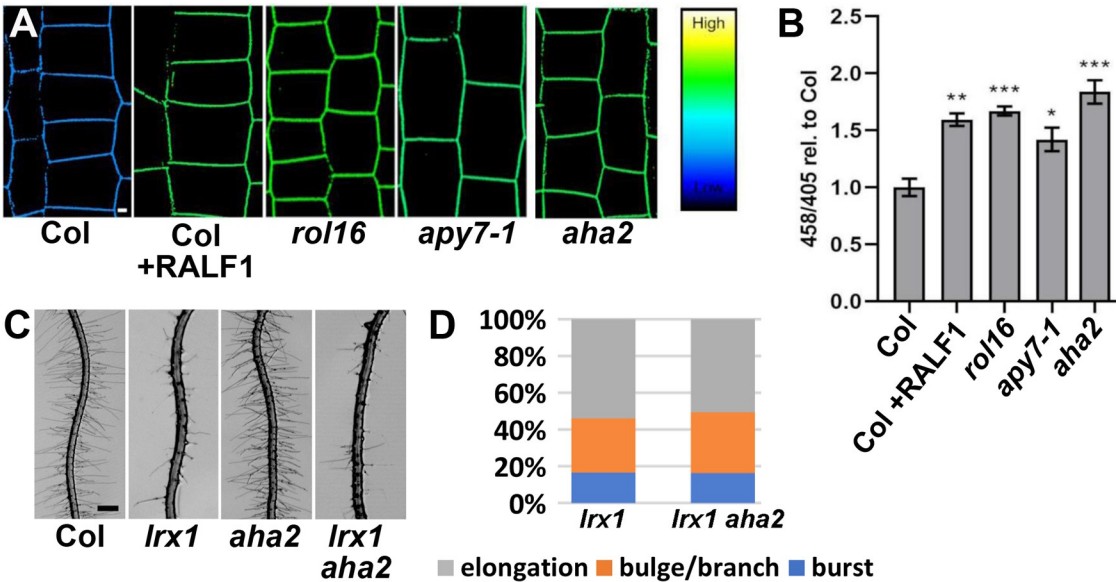

**Fig 4. The *rol16* mutants display more alkaline pH in the apoplast.** (A) Apoplastic pH was measured using HPTS staining in the root epidermal cells in seedlings grown for 4 days on half-strength MS media. Confocal imaging was performed to determine ratiometric pH change. The color code depicts intensity ratio 458/405 which correlates with pH values. The gradient on the right side represents reference colors for high and low values of pH. Col seedlings were treated with 1 μM RALF1 for 5 minutes and then immediately stained to detect the pH change. (B) Quantification of the apoplastic pH in mutants where the y-axis represents 458/405 values of the mutant roots relative to Col that was set to 1. The *rol16* mutant and *apy7-1* display an increase in pH which was significant compared to wild-type Col. The *aha2* mutant also shows more alkaline pH. The error bars represent STD. (student's t-test, n = 10, p *** < 0.0005, ** < 0.005, * < 0.05) (C) The mutation in *aha2* does not suppress the *lrx1* root hair phenotype. Bar = 500 μm. (D) Quantification of the three classes of root hair defects confirms that *aha2* does not impact the *lrx1* root hair defect. (student's t-test, n>100, p>0.5).

for acidification of the apoplast [41]. As expected, *aha2* seedlings revealed to have a more alkaline extracellular pH (Fig 4A and 4B). However, *aha2* failed to suppress the *lrx1* root hair defect (Fig 4C and 4D), indicating that the *lrx1*-suppressive effect of *rol16* or *apy7-1* is not solely based on the more alkaline apoplastic pH of these mutants.

## APY7 influences composition in cell wall polysaccharides

The biosynthesis of many cell wall-localized polysaccharides takes place in the Golgi [42]. Therefore, we investigated whether APY7 has an effect on cell wall structures which might contribute to the observed alterations in root (hair) development. To this end, seedlings were analyzed using a number of monoclonal antibodies (mAbs) against different cell wall epitopes (S2 Table), and fluorescence intensities of the FITC-labelled secondary antibody were quantified. The mAbs LM10 and LM15 binding xylan and xyloglucan, respectively [43,44], revealed differences in the labeling of wild-type and *rol16* mutant roots that were consistent over many samples. In the wild type, LM10 stained root hairs, and LM15 both the root and root hairs (Fig 5A and 5C), consistent with previous reports. LM15 labeling was stronger in the *rol16* mutant than the wild type, which was confirmed by quantification of the fluorescence (Fig 5C–5E). A qualitative difference was found for LM10 labeling, which was found only in wild-type but was undetectable in *rol16* mutant root hairs (Fig 5A and 5B). While the observation of differences in labeling intensities can also be explained by accessibility of the epitopes, these findings demonstrate that a mutation in *APY7* affects cell wall architecture, suggesting a role of the Golgi-localized APY7 in cell wall formation.

To analyze changes in cell wall composition in more detail, cell wall material of root tissue was collected, hydrolyzed, and neutral/acidic monosaccharides were quantified. As shown in Fig 5F, the quantities of sugars typically found in pectin and hemicelluloses were reduced in the *rol16* mutant compared to the wild type. While this finding does not conclusively answer whether the reduced xylose content reflects the reduced LM10-binding to the xylan, made of β-1,4 linked xylose [45], it is intriguing that sugars used for synthesis of Golgi-derived polysaccharides are overall reduced in the *rol16* mutant affected in the Golgi-localized APY7. Deficiencies in cell wall components such as cellulose can induce deposition of lignin [46], a possibility that was investigated. Seedlings as well as stem tissue were used for lignin staining, but no difference between the wild type and *rol16* mutant was observed (S4 Fig).

## APY7 abundance is influenced by ROS

FER is known to impact the amount of ROS [47,48]. If APY7 is a downstream component of a process involving FER, APY7 abundance might be influenced by ROS. To test this hypothesis, transgenic lines expressing *AYP7::APY7-GFP* were grown in the presence or absence of 5 μM hydrogen peroxide (for details, see Material and Methods) and GFP fluorescence was compared. In control conditions, APY7-GFP is detectable in the root tip but considerably stronger in older parts, i.e. in the mature zone of the root (Fig 6A and 6C). Reproducibly and with independent transgenic lines, treatment with hydrogen peroxide led to an increased abundance of GFP fluorescence in the root tip (Fig 6A and 6B) but a decrease in the adult root (Fig 6C and 6D), suggesting that APY7-GFP accumulates to higher levels in the tip region but is degraded in the mature root. Hence, the abundance of APY7 is differentially modified in a developmentally regulated manner. In a next step, it was investigated whether APY7 modifies ROS levels. To this end, wild-type and *rol16* mutant seedlings were grown and stained for ROS with Peroxy Orange 1 (PO1), a compound frequently used as a ROS stain. This analysis did not reveal a difference between ROS levels between the lines (Fig 6E). Hence, the influence of ROS on APY7 is unidirectional rather than being mutual.

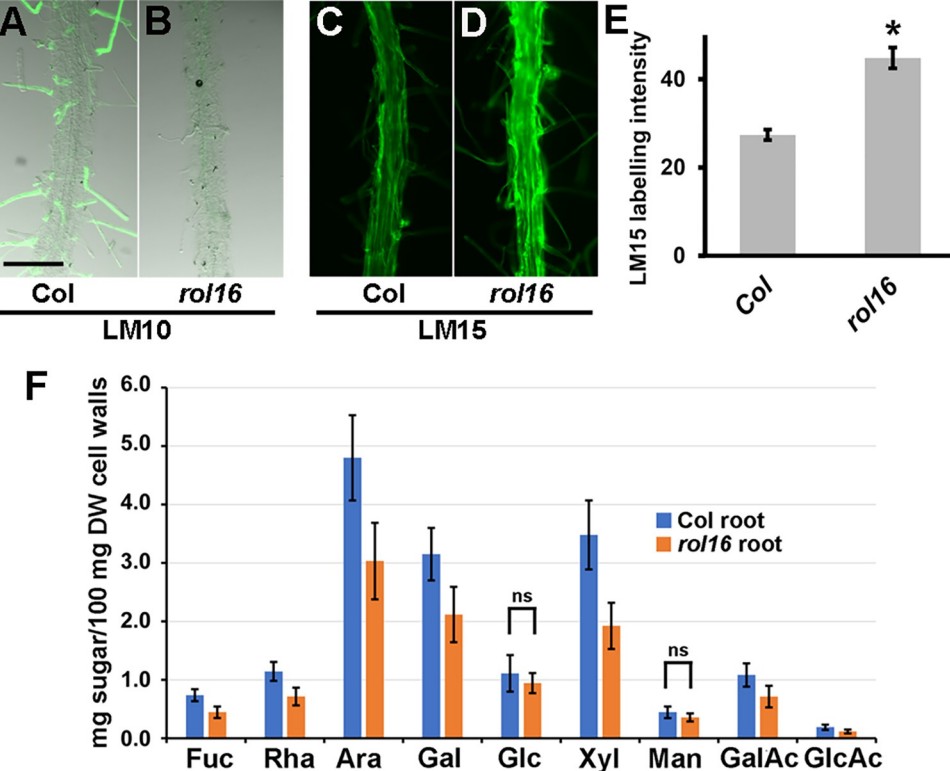

**Fig 5. Alterations in cell wall structures in the *rol16* mutant.** Six days-old seedlings were labelled with antibodies binding xylan (LM10, A,B) and xyloglucan (LM15, C-E) epitopes. (A,B) Combinations of fluorescence and transmission light microscopy to better visualize root (hair) structures that are only labelled in the wild type. (C,D) LM15 labeling along the main root is stronger in *rol16* compared to the wild type, which was quantified by ImageJ (E) (student's t-test, P<0.01, n>10). Bar = 0.5 mm. (F) Cell wall preparations (AIR: Alcohol Insoluble Residues) of root tissue of wild-type (Col) and *rol16* mutant seedlings were used for monosaccharide analysis. All differences are significant, except for the two labelled "ns". (n≥4, student's t-test, p<0.05).

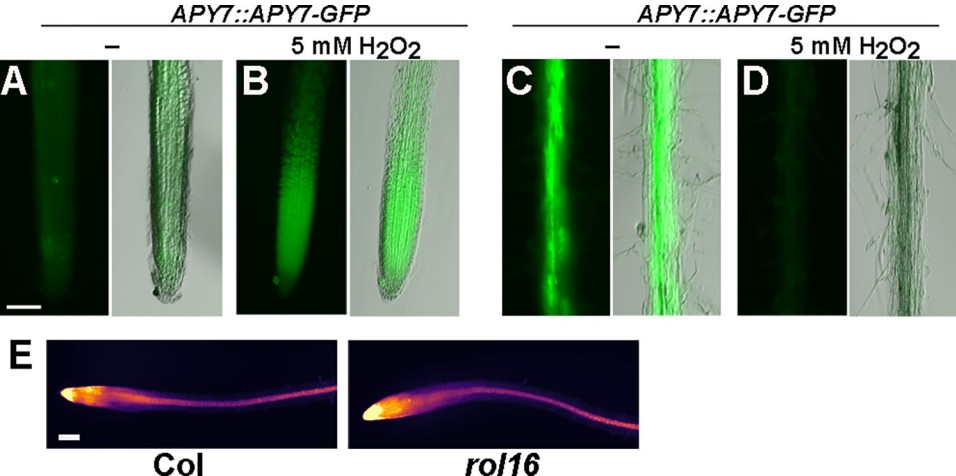

**Fig 6. APY7-GFP abundance is influenced by ROS.** (A-D) Seedlings were grown on MS plates in a vertical orientation for four days and then transferred to liquid MS for an additional two days prior to analyze GFP fluorescence. Representative pictures are shown in pairs, on the left with the fluorescence signal only and on the right an overlay with a bright-field image to clearly reveal the root structure. (E) Seedlings were grown for five days on MS plates and were then stained with the ROS stain PO1 (A-D, E) Bars = 300 μm.

### Increased sensitivity to exogenous ATP in *rol16* mutant

Apyrases can influence eATP levels as demonstrated for APY1 and APY2 [34,49]. Levels of eATP produced by Arabidopsis seedlings were measured after growth in liquid medium. Quantification of the eATP levels did not reveal a difference between the *rol16* and *apy7-1* mutants and the wild type (Fig 7A), suggesting that APY7 does not have an obvious impact on eATP levels. To examine whether APY7 is involved in responding to eATP, seedlings were grown on half-strength MS medium containing increasing concentrations of eATP, which induces root skewing [50] that can be quantified (S5 Fig). The quantification of skewing revealed a stronger response of the *rol16* and *apy7-1* mutants to increasing concentrations of eATP compared to the wild type (Fig 7B).

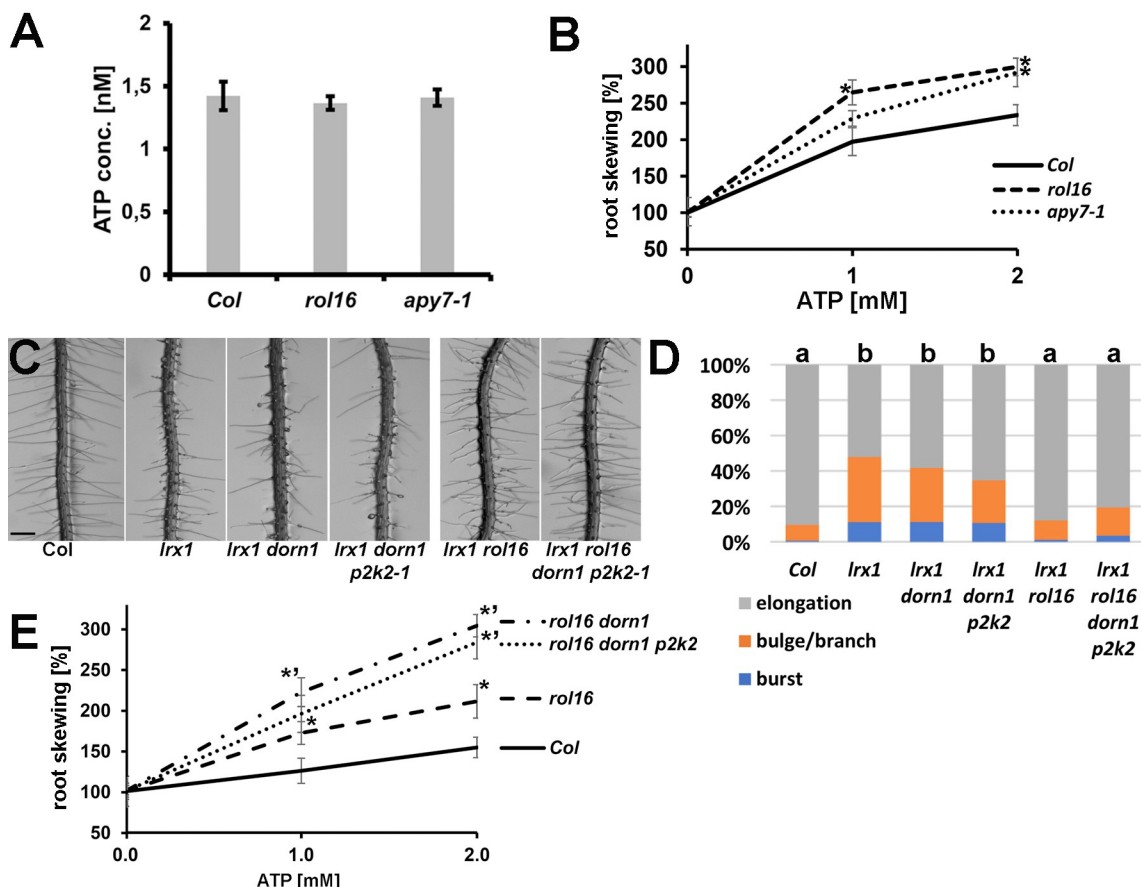

**Fig 7. Response to eATP and eATP receptors.** (A) Extracellular ATP (eATP) levels in seedlings are not altered in *apy7* mutants (n = 4, student's t-test p>0.1). (B) Sensitivity to eATP was measured by quantifying root skewing [74] relative to the value in the control sample with 0 mM ATP. The stronger deviation from the gravity vector, the higher the values (see also S5 Fig). The negative control (0 mM additional ATP) was set to 100% for each line. Asterisks indicate significant differences to the wild type (Col). The *apy7* mutants show a stronger effect upon ATP addition, resulting in increased skewing. (n>12, student's t-test, p<0.05). (C) Seedlings grown for six days in a vertical orientation are shown. Mutations in the ATP-receptor genes *DORN1* and *P2K2* do not influence the *lrx1* root hair phenotype nor do they influence the ability of *rol16* to suppress *lrx1*. Bar = 500 µm. (D) Classification of different types of root hair defects in the different lines shown in (C). Different letters indicate significant differences between the lines in all categories (n>250, student t-test p<0.01) (E) *dorn1* and *p2k2* mutations cause an augmented root skewing response to eATP. The negative control (0 mM additional ATP) was set to 100%. Significant differences (student's T-test, P<0.05; n > 8, error bars represent STD) between *rol16* and the wild type are indicated with asterisks (∗) and between *rol16* and *rol16 dorn1* or *rol16 dorn1 p2k2* with the asterisks (∗').

Hence, APY7 activity appears to mitigate the response of the root to eATP-induced growth stimulation. If APY7 is involved in the eATP sensing machinery, mutations in the two plasma membrane-localized ATP receptor proteins *P2K1/DORN1* and *P2K2* [29,30] would possibly also affect the *lrx1* root hair phenotype. An *lrx1 dorn1* double mutant was obtained by crossing the single mutants and showed the same defect in root hair development as the *lrx1* single mutant (Fig 7C and 7D). In a next step, the *lrx1 dorn1* double mutant was transformed with a *CRISPR/Cas9* construct targeting *P2K2*. Several lines with insertions of one or deletions of one to several base pair in *P2K2* could be retrieved (for details, see Material and Methods) which all changed the open reading frame, creating a stop codon within 30 amino acids (S6 Fig). Seedlings of these *lrx1 dorn1 p2k2* triple mutant lines also displayed the typical *lrx1* mutant root hair phenotype (Fig 7C and 7D). It was also tested whether DORN1 and P2K2-mediated perception of eATP is required for *rol16* -mediated suppression of *lrx1*. To this end, an *lrx1 rol16 dorn1 p2k2* quadruple mutant was established by crossing of the *lrx1 rol16* and *rol16 dorn1 p2k2* mutants. The *lrx1 rol16 dorn1 p2k2* mutant showed suppression of the *lrx1* mutant phenotype (Fig 7C and 7D), indicating that the effect of *rol16* as suppressor of *lrx1* does not depend on functional eATP perception at the plasma membrane.

Finally, eATP-induced root skewing was quantified in Col, *rol16*, *rol16 dorn1* and *rol16 dorn1 p2k2* lines. Here, the *rol16 dorn1* and *rol16 dorn1 p2k2* lines showed stronger root skewing than *rol16* (Fig 7E), suggesting that the two eATP receptor proteins also influence this effect. Together, these data suggest that the eATP receptors and APY7 function rather in parallel than in the same linear process to regulate eATP-induced plant development. This raises the question whether APY7 represents an alternative mechanism by which eATP is sensed at the plasma membrane. Arabidopsis proteome data found APY7 only in the Golgi [33] and this localization was confirmed by *APY7-GFP* expression in onion cells [32]. Here, a strongly expressing *APY7:APY7-GFP* line was used to investigate a possible co-localization by crossing with a line expressing the plasma membrane marker protein LTI6b-RFP. No evidence of a significant overlap of GFP and RFP fluorescence at the plasma membrane was observed (S7 Fig), confirming the previous findings [32,33]. This suggests that APY7 is involved in the intracellular response to eATP and that this response is independent of the signaling processes induced by the eATP receptors DORN1 and P2K2.

## Discussion

### APY7 is an unconventional apyrase protein

Apyrases degrade NTPs including ATP and thereby can influence many processes including cell growth [51]. Golgi-localized apyrases regulate levels of ATP required for glycosylations. ATP is also a signaling molecule that is excreted during stress conditions and this extracellular ATP (eATP) is in turn sensed by plants [27,28]. APY7 of Arabidopsis is a particular APY, since its structure with two transmembrane domains is reminiscent of ectoapyrases that localize to the plasma membrane. In Arabidopsis, only APY6 and APY7 have two transmembrane domains and APY7 has an additional and unique C-terminal domain [27]. The function of this domain and its influence on apyrase activity remains elusive. *APY7* is the only Arabidopsis *APY* not able to complement the yeast apyrase double mutant Δ*ynd1*Δ*gda1* [32], again pointing to a particular activity of APY7. This view is supported by the mutant phenotype of the two *apy7* alleles analyzed here, namely *rol16* and *apy7-1*, that show increased growth. In contrast, mutations in *APY1* and *APY2* inhibit growth and their overexpression causes increased growth [49,50,52]. While APY1 and APY2 negatively influence eATP levels [34], APY7 does not influence eATP levels but rather influences ATP sensitivity. Together, these findings reveal contrasting activities of APY7 compared to other Arabidopsis APYs.

## APY7 is involved in the intracellular response to eATP

In previous studies, application of anti-APY antibodies to pollen tubes increased eATP levels, likely as a consequence of antibody-mediated inhibition of APY activity. This proposes the existence of an APY at the plasma membrane with the enzymatic domain facing the apoplast that reduces eATP levels and is accessible for antibody binding [49]. APY7 is likely not this plasma membrane-localized APY as it was localized to the Golgi by an APY7-YFP fusion protein and was identified only in the Golgi proteome [32,33]. Finally, our analysis did not provide evidence for APY7 localizing to the plasma membrane. It cannot be excluded that small amounts of APY7 evading detection by microscopy are in the plasma membrane. Yet, this contribution would be minor considering that eATP levels appear unaffected in the *apy7* mutants. The increased sensitivity of *apy7* mutants to eATP suggests a role of APY7 in the sensing/ response to this signaling molecule. The two lectin-like receptor kinases DORN1/P2K1 and P2K2 are plasma membrane-localized ATP receptors that contribute to the eATP-induced $Ca^{2+}$ response and DORN1/P2K1 was shown to increase RBOHD-mediated ROS production [29,30]. Mutations in these RLKs reduce the sensitivity to eATP, yet neither affect the *lrx1* root hair defect in *lrx1 dorn1 p2k2* triple mutant nor do they prevent suppression of *lrx1* by *rol16* in an *lrx1 rol16 dorn1 p2k2* quadruple mutant. Hence, the effect of the *rol16* mutation appears independent of *DORN1* and *P2K2*. The increased response of the *rol16 dorn1 p2k2* triple mutant to eATP indicates that APY7 and DORN1/P2K2 likely work in parallel pathways. Hence, APY7 is rather responsible for the sensing or responding to intracellular ATP and is not activated via the eATP receptors at the plasma membrane.

## APY7 is part of the machinery downstream of the LRX/RALF/FER signaling module that alters cell wall composition

LRX1 and LRX2, as other LRXs, are high-affinity binding sites for RALF peptides and also function in conjunction with the FER receptor kinase [15–17]. Mutations in *APY7* alleviate to different degrees both the *lrx* and *fer* phenotypes, suggesting that APY7 activity is functionally connected to the LRX/RALF/FER signaling module. Likely, this does not involve a physical contact between APY7 and the other proteins since they are in different cellular compartments. A possible connection between the different proteins might function through ROS, the levels of which are influenced, among others, by FER [25,47,48], and which serve as signaling molecules involved in cell growth processes [53,54]. Our analysis revealed that ROS has an effect on APY7-GFP abundance, hence, FER potentially influences APY7 levels via ROS signaling. Further support for FER and APY7 being functionally related is provided by the observed synergistic effect between the *fer-4* and *rol16* mutant in respect to fertility. The strong reduction in frequency of homozygous *fer-4* mutants by *rol16* can be postulated to be a consequence of a combination of the *fer-4* mutant failing to induce pollen tube rupture at the synergid cells [9,39] and a more vigorous growth of mutant pollen tubes that are less prone to burst upon perception at the synergid.

RALF1 is bound by LRX1 and LRX2 [17] as well as FER [13] and induces alkalinization of the growth medium and a reduction in cell growth [5]. The reduced sensitivity of the *apy7* alleles to RALF1 is consistent with APY7 functioning downstream of the LRX/RALF/FER signaling module. Mutations in *APY7* as well as a reduced *RALF1* expression both result in an increased cell elongation (data shown here and [55]), correlating with the proposed function of RALF1 and APY7 in limiting cell growth. A possible explanation for the observed reduction in sensitivity towards RALF1 might be that the *apy7* mutants show an increased apoplastic pH, which in wild-type seedlings is caused by a RALF1-induced reduction in AHA-type proton exporters [56]. This observed alkalinization of the extracellular matrix of *apy7* mutants,

however, is not the cause of suppression of *lrx1*, as *aha2* mutants also show a more alkaline apoplast [56] but do not modify the *lrx1* mutant phenotype. The less acidic pH of *apy7* mutants also does not follow the "acid growth hypothesis", according to which growth correlates with acidification of the apoplast [57]. This hypothesis, however, is also challenged, for example by the finding that along the root axis, the pH does not strictly correlate with cell expansion and the lowest pH is found in the transition zone containing non-elongating cells [58]. Expansins are involved in cell wall expansion and function non-enzymatically, likely by temporarily detaching cellulose-hemicellulose connections [59]. Interestingly, the two antibodies consistently detecting alterations in cell wall architecture of *rol16* compared to the wild type are directed against hemicelluloses, namely Xyloglucan (XG) and Xylan. These changes in cell wall structures might influence the expansion potential of the cell walls. XG is postulated to interconnect cellulose microfibrils by establishing "hotspots" on cellulose-XG interactions [60] and root hairs with reduced xyloglucan content are shorter than those of the wild type [61]. Also, previous work revealed that RALF1 influences the level of *TCH4* expression [55], a gene encoding a xyloglucan-endo-transglycosylase that promotes cell (wall) elongation [62,63]. It is conceivable that the increased XG levels observed in the *rol16* mutant might have an opposite effect resulting in the observed increased cell elongation. Xylan is also a hemicellulose found in all cell walls but particularly abundant in secondary cell walls [64,65]. Opposite to XG, xylan detection is strongly reduced in the *rol16* mutant. Whether the absence of xylan is compensated for by XG or whether the metabolic flow of xylose is simply redirected to XG biosynthesis remains to be determined. The overall sugar composition of *rol16* mutant root cell wall material suggests a general decrease in the content of neutral and acidic sugars forming pectin and hemicelluloses compared to the wild type. It is noteworthy that these polymers are synthesized in the Golgi apparatus [66], raising the question whether APY7 is involved in maintaining the level of their biosynthesis. Lignin deposition is increased likely as a compensatory effect under cellulose deficiency [46], but also in lines with reduced levels of APY1 and APY2 [34]. Both effects are not observed in *rol16*, again suggesting that APY7 functions in a different way than other apyrases. There are certainly other factors in the cell wall that influence its expansion. For one, enzymes modulating hemicellulose networks such as XG-hydrolases, but also PMEs (pectin methylesterases), some of which are also active in a slightly alkaline pH [67,68]. PME activity can be a signal to trigger pectin turnover and cause cell elongation. Cell wall architecture is a key determinant of cell growth, influencing the physical properties of the cell wall and limiting its expansion. Even if the changes in antibody labeling in the *rol16* mutant are due to changes in accessibility rather than the abundance of the epitopes, they demonstrate a modification of the cell wall in the absence of APY7. The biosynthesis of several cell wall polymers takes place in the Golgi, where APY7 is predominantly localized, suggesting that the initial biosynthetic steps can influence the final cell wall architecture. It remains to be shown exactly how APY7 influences cell wall composition, whether this is via regulating the import of the large variety of sugar monomers required for polymerization, regulating levels of ATP (and other NTPs) in the Golgi lumen which in turn influences enzyme activity, or by affecting vesicle transport, to name three of many possibilities. The differing phenotypes of plants with altered APY1, APY2, and APY7 levels strongly suggest that apyrases show considerable differences in their activities, resulting in very different output effects on growth processes.

## Materials and methods

### Plant material, growth, EMS mutagenesis and cloning

EMS mutagenesis on *lrx1* loss-of-function mutant has been previously described [69]. The mutant lines used for mapping *ROL16* have the Columbia (Col) genetic background. Seeds

were surface sterilized by a solution of 1% sodium hypochlorite, 0.03% Triton X-100, washed three times with distilled water, plated on half-strength MS (2% sucrose, 0.6% gelrite; unless stated otherwise), and stratified for 2–3 days at 4˚C. The plates were then transferred to the growth chamber with a 16h-light/8h-dark cycle at 22˚C and grown vertically. Seedlings were transferred to soil for crosses and propagation, and grown in a growth chamber with the 16h-light/8h-dark cycle at 22˚C. For genotyping, the molecular marker for *lrx1* has been described [69]. For *rol16*, the primers SG23 and SG24 (all primers are listed in S1 Table) produce a PCR fragment with a *Spe*I site in *rol16* but not the wild type. The *apy7-1* mutant was obtained from NASC (SALK_124009) and the T-DNA insertion was detected with the primers LbB1 and SG76, the wild-type copy with SG75 and SG76. The *dorn1* mutant was obtained from NASC (SALK_042209) and detected with the primers LbB1 and *dorn1_R*, the wild type with *dorn1_F* and *dorn1_R*.

For complementation analysis, a 1.5 kb of *APY7* promoter was amplified with *APY7-Prom_F/APY7Prom_R*, digested with SpeI and AscI, and cloned into *pGPTV-Kan* [70] where the *GUS* gene was cut out with AscI/SacI and replaced by *GFP* amplified from *pMDC99* [71] with *GFP_AscI_F* and *GFP_SacI_R*. The *APY7* CDS was amplified with *APY7_F* and *APY7_R*, digested with AscI and ligated into the *APY7::GFP* clone digested with AscI to obtain *APY7::APY7-GFP*. For the *CRISPR/Cas9* mutagenesis, the *pKI1.1* plasmid [72] digested with AarI was ligated with the *P2K2*-specific double-stranded oligo designed based on CHOPCHOP [73] using the primers P2K2_KI1_F/ P2K2_KI1_R. For identification of a *rol16* single mutant, the *lrx1 rol16* line was crossed with wild-type Columbia and seedlings of the F2 population were selected with the established markers for the genotype $LRX1^{+/+}rol16^{-/-}$.

## Plant transformation and *CRISPR/CAS9*-mediated mutagenesis in P2K2

Plant transformation was done by floral dipping using Agrobacterium GV3101. T1 seeds of plant transformed with *APY7::APY7-GFP* in the binary vector *pGPTV-Kan* were selected on kanamycin. Several independent T1 transgenic lines were selected and propagated. T1 seeds of plants transformed with the *P2K2*-specific gRNA in the *CRISPR/Cas9* vector *pKI1.1* were selected based on the RFP fluorescence. Cauline leaves of inflorescences of different T1 plants were used to extract DNA and detect InDels in *P2K2* by PCR amplification of the target region with the primers P2K2_F1 and P2K2_R1, followed by sequencing with P2K2_F1.

## ROS treatment

For ROS treatment, independent APY7::APY7-GFP transgenic lines were grown for four days on half-strength MS plates and then transferred to liquid MS (half-strength MS, 1% sucrose) for an additional two days. For the treatment, 5 µM hydrogen peroxide (Sigma) was added to the sterilized medium. Plants were kept in six-well plates (Greiner Bio One) in the growth chamber mentioned above.

## Microscopic analysis

Root hair phenotypes were determined with seedlings grown in a vertical orientation for six days, using an MZ125 stereomicroscope (Leica) and images were obtained with a DFC420 digital camera (Leica). *APY7-GFP* localization was done on a Leica SP5 confocal microscope (Leica) equipped with an Argon laser (488 nm) and Diode-Pumped Solid-State (DPSS) laser (561 nm), hybrid detectors and a 63x (N.A. 1.40) oil immersion objective. Fluorescence emissions were filtered between 500 nm and 550 nm for GFP and 580 nm and 700 nm for RFP. In order to obtain quantitative and comparable data, experiments were performed using strictly identical confocal acquisition parameters (e.g. laser power, gain, zoom factor, resolution and

emission wavelengths reception). The parameters were selected for low background and no pixel saturation. Intensity measurements and intensity plots were analyzed using ImageJ. For the detection of APY7-GFP fluorescence upon ROS treatment, Seedlings were placed on a glass slide containing AF1 anti-fade (Citifluor), covered with a cover glass and fluorescence was analyzed with a thunder microscope (Leica) under non-saturating conditions using a eGFP filter. Parameters of the pictures were kept strictly identical among samples to be tested.

## GUS staining

Seedlings of several T2 families representing independent transformation events containing the *APY7::GUS* construct were grown and stained for GUS activity. To this end, seedlings were vacuum infiltrated in 0.5 mg/ml X-Gluc, 50 mM Na-phosphate pH 7, 10 mM EDTA, 0.1% Triton X-100 and incubated at 37˚C for several hours. GUS staining was subsequently stopped by exchanging the incubation buffer by 70% EtOH. Pictures were taken by a stereomicroscope (Leica).

## ROS detection

Hydrogen peroxide was visualized with PO1. PO1 (Tocris) was dissolved in DMSO to make a 500 μM stock and was further diluted in water to make a 40 μM working solution. Seven days-old seedlings were incubated in PO1 for 5 min in the dark and were then rinsed with water and mounted in water for imaging. Seedlings were imaged on a Leica M205 FCA fluorescence stereo microscope using a 545-620nm filter.

## RNA extraction and quantitative RT-PCR

Dynabeads mRNA DIRECT Kit 61011 and 61012 (61021) from ThermoFisher was used to extract mRNA directly from grinded leaf tissue. 3μL of bead-bound mRNA was reverse transcribed using the iScript advanced kit (BioRad). qRT-PCR was performed using the primers qRT_apy7_5'_F, qRT_apy7_5'_R, qRT_apy7_3'_F and qRT_apy7_3'_R (S1 Table). The expression was normalized against the housekeeping genes *ACT2*, *EFα* and *UBQ10* using the following primer pairs: ACT2_F and ACT2_R, EFα_F and EFα_R, UBQ10_F and UBQ10_R (S1 Table).

## Immunoblotting

Seedlings were grown for 10 days in a vertical orientation on half-strength MS plates, and 100 seedlings were collected and grinded in liquid nitrogen. Proteins were extracted with 200 μl 1% SDS, boiled for 5 minutes, cooled on ice, centrifuged, and 20 μl were mixed with 5 μl 5x Lämmli buffer for loading on a 10% SDS-PAGE gel (BioRad). After blotting was done on nitrocellulose using the semi-dry system (BioRad), the membrane was blocked overnight with 1xTBST, 5% non-fat dry milk powder. Immunolabeling was done in 1xTBST, 1% non-fat dry milk powder with a 1:3'000 dilution of an anti-GFP antibody (Biolegend, 902601), followed by a 1:5'000 dilution of an anti-mouse-HRP antibody (Sigma Aldrich, A 4416).

## Cell wall analysis by immunolabeling and lignin staining

Seedlings were grown in a vertical orientation on the growth medium described above, and incubated in 1x PBS, 3% non-fat milk powder, for 1 h with 10-fold diluted anti-cell wall epitope mAbs and 100-fold diluted secondary, FITC-labelled anti-rat antibody (Sigma F1763). Each antibody incubation was followed by washing three times for 10 min each with 1x PBS. Seedlings were placed on a glass slide containing AF1 anti-fade (Citifluor), covered with a cover glass and fluorescence was analyzed with a thunder microscope (Leica) under non-

saturating conditions using a eGFP filter. FITC labeling was quantified by ImageJ and the values for the wild type were set to 1. The microscope and camera settings were maintained for the same mAb, but different mAb required different settings, thus the quantifications of different mAb cannot be compared. Each antibody labeling was done with several seedlings and repeated at least five times. Only consistently observed differences in mAb labeling were considered reliable.

Wiesner staining was used to reveal lignin deposition in seedlings and stem sections. To this end, 0.3 g of Phloroglucinol was dissolved in 10 mL 100% ethanol and 5 mL fuming HCl (36% v/v) was added. The plant material was incubated for 15 minutes and then analyzed with a stereomicroscope (Leica).

## Growth inhibition assay by RALF1 treatment

The seedlings were grown for three days in a vertical orientation on half-strength MS plates with 1% sucrose and 1% bactoagar. After that, they were transferred to half-strength MS (1% sucrose) liquid medium containing 1 μM RALF1 peptide and grown for three more days. The seedlings were laid on an agar plate and root length was measured using Fiji ImageJ software. For cell size measurements, the seedlings were grown vertically on half-strength MS plates for 6 days. The epidermal cells in the differentiation zone of primary roots were imaged using a Zeiss Axio Imager Z1 microscope. The cell length was measured using Fiji ImageJ software.

## Extracellular ATP analysis

For ATP sensitivity tests, the seeds were plated on half-strength MS plates (0.6% gelrite) supplemented with 0 mM, 1 mM ATP, 2 mM ATP and 4 mM ATP. Seedlings were grown vertically for six days, scanned, and the root length and progression on the gravity vector were measured using Fiji ImageJ software to determine the root skewing index [74].

The ATP quantification was performed using ENLITEN ATP Assay System Bioluminiscence Detection Kit from Promega (protocol by [50]). The seedlings were grown in half-strength MS liquid medium for five days and the MS medium was collected. A standard curve was produced with the ATP standard provided in the kit which was used to calculate ATP concentration. Five biological replicates per genotype were used.

## Extracellular pH analysis

The protocol for using HPTS dye has been described in [40]. The seedlings were grown vertically on half-strength MS media for 4 days. Half-strength MS media containing 1 mM HPTS (Sigma-Aldrich) plates were also prepared for confocal imaging in borosilicate imaging chambers (Lab Tek). The seedlings were stained with 1 mM HPTS dye and immediately used for confocal imaging in the Leica SP5 microscope. Fluorescence was collected for two forms of HPTS using two different channels: protonated (Excitation at 405 nm and emission peak at 514 nm) and deprotonated (Excitation at 458 nm and emission peak at 514 nm).

A 63x glycerol immersion objective lens was used and the fluorescence data collected was analyzed using Fiji ImageJ software. Ten biological replicates per genotype were used for the experiments. Ratiometric image conversion was performed using the Fiji script that was customized by [40]. Statistical analysis was done using GraphPad Prism8 software.

## Neutral and acidic sugars composition analysis of cell walls

For the non-cellulosic monosaccharide composition analysis, 5 to 6 mg of freeze-dried, 10 days-old roots from Col and *rol16* were used for cell wall extraction, as described in [75].

Following digestion with α-amylase (from Bacillus species, Sigma) according to [76], 1 to 2 mg of destarched dry cell wall (DCW) was hydrolyzed with 2 N trifluroacetic acid (TFA) for 90 min at 121˚C. After evaporation of TFA under nitrogen stream, samples were dissolved in 1 mL $H_2O$ and 2.5 µL of the sample was injected onto a Dionex CarboPac PA1 column (2mm ID x 250mm) with a high-performance anion exchange chromatography (HPAEC DIONEX ICS5000+) and sugars were detected using pulsed amperometry (PAD) with the Dionex Gold Carbohydrates Quad potential. Separation phases for neutral and acidic sugars was according to the protocol described in [77] and data were collected and processed using the Chromeleon 7.2.10 software (Thermofisher Scientific Inc.). Analyzes were performed for 4 to 5 biological replicates and results are expressed in mg.100mg$^{-1}$ DCW. For quantification purposes, standard curves for each of the sugars were realized using stock concentrations (L-fucose/25 µg. mL$^{-1}$, L-rhamnose/105 µg.mL$^{-1}$, L-arabinose/105 µg.mL$^{-1}$, D-galactose/150 µg.mL$^{-1}$, D-glucose/100 µg.mL$^{-1}$, D-xylose/100 µg.mL$^{-1}$, D-mannose/50 µg.mL$^{-1}$, D-galacturonic/150 µg. mL$^{-1}$, D-glucuronic acid/50 µg.mL$^{-1}$) and one-half, one quarter and one-tenth dilutions.

## Supporting information

**S1 Fig. Mutation in *APY7* and comparison to related APYs. (A)** List of SNPs in coding sequences around *ROL16* (At4g19180, indicated in bold), as obtained from WGS. **(B)** The C to T mutation in *ROL16* was confirmed by sequencing. As the coding strand orientation is opposite of the WGS data, it is a G to A polymorphism (indicated with black arrow). A CAPS marker (for details, see Material and Methods) was established for simple detection. **(C)** Upper panel; schematic drawing of the APY7 protein and genomic DNA, with the protein coding sequence (indicated in grey) interrupted by an intron. qRT-PCR was performed with two primer pairs, located in the 5' and 3' UTRs, respectively. Total RNA was extracted from wild type (Col), *lrx1*, *lrx1 rol16*, and *lrx1 apy7-1* mutant seedlings. The lower panel shows quantification of RNA levels with the wild type set to 1. **(D)** Comparison of APY7 with APY1,2,6 of Arabidopsis and a human NTPDase. APY6 is the only Arabidopsis APY, besides APY7, to have two TMDs. APYs have five conserved motifs (apyrase conserved regions, ACRs), in APY7 corresponding to positions 150–157 (ACR1), 236–246 (ACR2), 281–293 (ACR3), 312–319 (ACR4), and 566–569 (ACR5). The amino acid sequences of the ACRs are listed. APY7 has a C-terminal extension not found in any other APY protein in Arabidopsis, the biological significance of which remains elusive. (DOCX)

**S2 Fig. Expression of *APY7* in roots of *Arabidopsis thaliana*. (A-D)** *APY7::GUS* fusion construct transformed into Col-0 for expression analysis. The 6-days-old seedlings were stained with GUS for 3–4 hours and the reaction was stopped by adding 70% ethanol. *APY7::GUS* activity was observed in the root tissue. GUS staining was observed at the lateral root initiation site **(A)**, in the vasculature of the maturation/ differentiation zone **(B)**, diffused expression in the elongation or cell division zone **(C)**, and high expression in the root tip **(D)**. **(E)** GFP fluorescence induced by an *APY7::APY7-GFP* construct reveals expression in all cells with a punctate structure likely representing the Golgi. Bar = 500 µm (A-D have the same magnification). (DOCX)

**S3 Fig. Interaction of the *rol16* and *fer* mutations. (A)** *rol16* alleviates the root hair developmental defect induced by the *fer-4* knock-out mutant, with more root hairs successfully entering the elongation phase, while remaining shorter than in the wild type. Bar = 300 µm. Classification of different types of root hair defects in the wild type (Col) compared to *fer-4* and *fer-4 rol16*. Different letters indicate significant differences between the lines (student's t-test, n>250, p<0.001) **(B)** The frequency of homozygous mutant *fer-4* is strongly reduced by

five-fold in the *rol16* mutant background compared to a wild-type *ROL16* background, suggesting a genetic interaction between *rol16* and *fer-4*. **(C)** Siliques of *fer-4 rol16* double mutants are shorter and contain less seed than the respective single mutants. Bar = 1cm.
(DOCX)

**S4 Fig. Comparable lignification of wild type and *rol16*.** Seedlings and stem sections were incubated in Wiesner staining solution. **(A)** Seedling hypocotyls, **(B)** seedling roots, **(C)** stem sections. Both distribution and intensity of lignification are comparable between the wild type and the *rol16* mutant plants. Bars = 300 μm (A), 200 μm (B), 1 mm (C).
(DOCX)

**S5 Fig. Quantification of root skewing.** Schematic representation on the left: the ratio of the total length of the seedlings root (L) over the progression on the Y-axis (ΔY) describe the arccosinus of Θ, a mathematical description of growth behaviour [74]. Stronger root scewing results in an increase in angle Θ. Right: example of Col versus *rol16* mutant seedlings grown with 0 mM ATP (top) and 1 mM ATP (bottom), used for data acquisition. It reveals that with eATP treatment, Θ values are bigger in *rol16* mutants compared ot the wild type.
(DOCX)

**S6 Fig. DORN1 and P2K2 protein structure. (A)** *P2K1/DORN1* (At5g60300) and *P2K2* (At3g45430) encode lectin-like receptor kinases that bind ATP. Numbers indicate amino acid positions, the mutants being used in this work are indicated, the *dorn1* mutant is the T-DNA insertion line SALK_042209 with the insertion site corresponding to amino acid codon 92, *p2k2* represents *CRISPR/CAS9* -induced mutations at the codon 107 that change the reading frame and terminate translation after around 30 amino acids. Numbers indicate amino acid positions. **(B)** *P2K2* expression levels were determined by qRT-PCR on RNA samples from wild-type Col and different *p2k2* alleles, with Col arbitrarily set to 1.
(DOCX)

**S7 Fig. APY7 is not a genuine plasma membrane protein.** Plants expressing APY7-GFP with the plasma membrane marker LTI6b-RFP were used for colocalization. A virtual section through cells revealed strong RFP signal at the cell periphery due to LTI6b-RFP fluorescence at the plasma membrane. By contrast, no signal beyond background could be identified with the GFP filter identifying APY7-GFP.
(DOCX)

**S1 Table. Primers used in this study.** Positions in bold (SG23) indicates SNP compared to genomic DNA to create a CAPS marker with a SpeI in *rol16*.
(DOCX)

**S2 Table. Monoclonal antibodies used in this work.** A list of these antibodies, their specificities and list of publications describing them can be found at https://plantcellwalls.leeds.ac.uk/wp-content/uploads/sites/103/2021/11/JPKab2021.pdf
(DOCX)

## Acknowledgments

We acknowledge the technical assistance of Solène Bassard in preparing the samples for cell wall analysis.

## Author Contributions

**Conceptualization:** Jérôme Pelloux.

**Data curation:** Shibu Gupta, Amandine Guérin, Aline Herger, Xiaoyu Hou, Myriam Schaufelberger, Romain Roulard, Anouck Diet, Valérie Lefebvre, Christoph Ringli.

**Formal analysis:** Amandine Guérin, Aline Herger, Xiaoyu Hou, Myriam Schaufelberger, Romain Roulard, Anouck Diet, Stefan Roffler, Jérôme Pelloux.

**Funding acquisition:** Christoph Ringli.

**Investigation:** Shibu Gupta, Aline Herger, Christoph Ringli.

**Methodology:** Shibu Gupta, Aline Herger, Romain Roulard, Stefan Roffler, Valérie Lefebvre, Jérôme Pelloux.

**Project administration:** Christoph Ringli.

**Resources:** Valérie Lefebvre.

**Supervision:** Thomas Wicker, Jérôme Pelloux, Christoph Ringli.

**Visualization:** Shibu Gupta, Aline Herger, Xiaoyu Hou.

**Writing – original draft:** Thomas Wicker, Jérôme Pelloux, Christoph Ringli.

**Writing – review & editing:** Shibu Gupta, Amandine Guérin, Aline Herger, Xiaoyu Hou, Myriam Schaufelberger, Anouck Diet, Stefan Roffler, Thomas Wicker, Jérôme Pelloux, Christoph Ringli.

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
