## [Decision Letter · Decision Letter 0]

28 May 2023

Dear Dr Ringli,

Thank you very much for submitting your Research Article entitled 'Growth-inhibiting effects of the unconventional plant APYRASE 7 of Arabidopsis thaliana influences the LRX1/FER/RALF growth regulatory module' to PLOS Genetics.

The manuscript was fully evaluated at the editorial level and by two independent peer reviewers. The reviewers appreciated the attention to an important problem, but raised some substantial concerns about the current manuscript. Based on the reviews, we will not be able to accept this version of the manuscript. However, we would be willing to review a much-revised version. We cannot, of course, promise publication at that time.

If you decide to revise the manuscript for further consideration at PLOS Genetics, please aim to resubmit within the next 60 days, unless it will take extra time to address the concerns of the reviewers, in which case we would appreciate an expected resubmission date by email to plosgenetics@plos.org.

We are sorry that we cannot be more positive about your manuscript at this stage. Please do not hesitate to contact us if you have any concerns or questions.

Yours sincerely,

Li-Jia Qu

Section Editor

PLOS Genetics

Reviewer's Responses to Questions

**Comments to the Authors:**

Reviewer #1: The manuscript by Gupta et al. describes a relationship between APYRASE7 (APY7) and LRX1/FER/RALF module. By using genetics approaches, the authors observed that APY7 suppress the growth defects in lrx and fer mutants. Further, they found that APY7 modulates the growth-inhibiting effect of RALF1, influences the cell wall components, and alters apoplastic pH. This manuscript focused on an interesting topic. However, there are quite a few critical comments/concerns that should be adequately addressed. Especially, how APY7 spatially inhibits LRX1/FER/RALF? how to explain the cell elongate faster with a more alkali apoplastic pH in apy7-1? I suggest the authors address the following comments and significantly strengthen this work by additional data.

Major

1. APY7, LRX and FER have different subcellular localization patterns. It is hard to imagine how Golgi apparatus-localized APY7 spatially work with cell surface-localized RALF1/LRX/FER.

2. I am not convinced that eATP induces the gravity-dependent root skewing of rol16. Because root skewing does not mean that plant do not have gravitropism. It would be necessary to observe root growth using a soil-based method.

3. In Fig1A, APY7::APY7-GFP is complemented into the lrx rol16 double mutant background. It would be better to select at least two lines to observe the root hair phenotype. Besides, can different abundances of APY7 restore the defect in root hair development at different levels?

4. An additional experiment of APY7::APY-GFP in fer-4 apy7-1 should be added in Fig 3C to demonstrate that APY7 has a function in growth processes that are influenced by RALF1.

5. The peak map or raw data of GWS and PCR gel images of SNP molecular markers should be provided in supplementary data.

6. Except for genetics, biochemical data are needed to support the conclusion that APY7 inhibits the LRX/RALF/FER pathway, such as interactions-related experiments using BiFC and Co-IP.

7. Further experiment could be carried out to characterize the APY7-dependent changes in structures and functions of the cell wall. For instance, chemistry approaches can be used to quantify the content and composition of cellulose, hemicellulose, and lignin in apy 7-1 mutant, electron microscopy to observe the ultrastructure and morphology of its cell wall.

8. The phenotype of root hairs in the picture needs corresponding index data statistics, such as root hair density and number of root hairs.

9. In the abstract and introduction, the authors mentioned that cell wall formation id modulated by LRX/RALF/FER complex. But the related reference is missing.

10. How to explain the root cell of apy7-1 elongate faster with a more alkali apoplastic pH?

11. The mechanism of apy7-1 affecting the cell wall components through RALF1/LRX/FER is not clear.

 

Minor

1. In Abstract, “LRX/RALF/FER” modules should be written in a same order.

2. Line # 44: “resuling” is miswritten?

3. Lines 123-125: The author should demonstrate the segregation ratio of F2 generation: how many lines were restored or not?

4. Lines 149 and 374. “AYP7” and “AYP1” are miswritten?

5. Error in Line 233, 1 uM RALF1. Line 257 1 μM RALF1. Should be unified.

6. Line 264. An error in the unit of scale bar.

7. Line 563: An error in “in the Leica SP5 microscope”.

8. Line 568: Five biological replicates are not enough for HPTS assay.

9. Line 336: I suggest replace “cell wall structures” with “cell wall components”.

10. The author compared the expression levels of APY7-GFP among different T2 lines through Western blot assay in Fig 1C. How to ensure that the loading amount is the same for each line? Please clarify it.

11. In Fig S2E, wild-type seedling should be added as a control under the same parameters to avoid the auto-flourescence. Besides, the value scale bar of this figure is missing. “ABCDE” font sizes are not uniformed.

12. In Fig4A, the position of apy7-1 root cell seems different from those of other samples.

13. GFP intensity in Fig S2E and Fig S6 is too low to be convinced. One possible experiment is to investigate subcellular localization of APY7 and expression patterns in response to different concentrations of eATP.

14. The authors mentioned RALF 23 and RALF 33 in the method (Line 544). But no related data was presented in the manuscript.

15. It is counterintuitive that RALF1 can promote root growth in fer-4 in Fig 3C. Why is there a huge difference in root lengths between Fig 3A and 3C, if the plants growth under similar conditions?

16. Fig 4 shows an increase in cell wall pH values for the rol16 and apy7-1 mutants, indicating that APY7 affects the cell wall pH in root tissues. The authors should also present the apoplastic pH of fer5 rol16.

17. The authors analyzed the composition of the cell wall in seedlings using multiple monoclonal antibodies in Fig 7. Were the immunological efficiencies consistent for different samples?

18. There are many elements missed in the figures: statistical analysis in FigS1 A and Fig 5A; phenotypic image in Fig 3; scale bar in Fig S3C, Fig S4 and Fig 6; unit in Fig S6B; confocal image in Fig S6. Some of them are really important for the audience.

Reviewer #2: In this manuscript, the authors tested quite a few hypothesis, trying to decipher how LRX, FER, APY7 may involved in root hair formation. From my point of view, this manuscript is still preliminary, still need much more investigation.

Major points:

1. Fig 1 showed root hair defect in lrx1 mutant, the introduction of rol6 repressed root hair defect in lrx1 mutant, Fig 2 and S3A Fig showed similarity of lrx1/lrx2 to fer-4/fer-5, the introduction of rol6 repressed root hair defect in fer-5 mutant, but only partially repressed root hair defect in fer-4 background. This suggest that shorter root hairs in lrx1 may be similar to that in fer-5, but root hair defects (may be root hair burst) more severe than the shorter root hair in fer-4 are not repressed by rol6, which also suggest that root hair burst defect in fer-4 may not be correlated with APY7. The authors need to provide clear evidence showing different types of root hair defect in different mutants.

2. Line 200-205, also in S3C Fig, the authors reported that rol16 boosted fer-4 defect in seed set. Together with their observation that rol6 only repressed the short root hair defect in fer-4, these data suggest that ROL6 may only function in ROS signaling pathway. The authors need to test whether lrx1 also have fertilization defect.

3. Line 215-270 (Fig 3, Fig 4) reported unrelated data, it is confusing to the readers why this is included. Why not directly describe Apyrase function starting from Line 272 (Fig 5)?

4. Line 272-334 (Fig 5, Fig 6) reported the possible involvement of eATP and receptor proteins P2K1/DORN1 and P2K2. Again, this is not involved. Obviously, the gravitropic response caused by eATP is not correlated with root hair defect. Suggest to remove this data.

5. Finally, based on staining of cell wall xylan and xyloglucan, the authors concluded APY7 influences cell wall structures, but they did not correlate this with cell wall defect. They also did not stain other root hair mutant with LM10 and LM15.

Minor points:

1. Pay attention to typo errors, such line:4-45：”resuling” in cells that are malformed, “to” small or big. “resuling” should be “resulting”, “to” should be “too”

2. Line 123-125, “Backcrossing of the lrx1 rol16 mutant with lrx1 resulted in the F2 generation in a 3 : 1 segregation of lrx1 : wild type-like root hair formation, indicating that the rol16 mutation is recessive.” The conclusion from this experiment need to provide data and point to the corresponding supplementary data.

3. Line 196 showing in a rol16 mutant population segregating for fer-4 (rol16 -/- fer-4 +/-), however, only 12 out of 800 seedlings were homozygous for fer-4. But in S3B Fig, the sample name is written as rol16+/+; fer-4 +/-, should be corrected to rol16 -/- fer-4 +/-.

**Have all data underlying the figures and results presented in the manuscript been provided?**

Reviewer #1: Yes

Reviewer #2: Yes

PLOS authors have the option to publish the peer review history of their article (what does this mean?). If published, this will include your full peer review and any attached files.

Reviewer #1: No

Reviewer #2: No

---

## [Decision Letter · Decision Letter 1]

29 Nov 2023

Dear Dr Ringli,

We are pleased to inform you that your manuscript entitled "Growth-inhibiting effects of the unconventional plant APYRASE 7 of Arabidopsis thaliana influences the LRX/RALF/FER growth regulatory module" has been editorially accepted for publication in PLOS Genetics. Congratulations!

Yours sincerely,

Li-Jia Qu

Section Editor

PLOS Genetics

Comments from the reviewers (if applicable):

Reviewer's Responses to Questions

**Comments to the Authors:**

Reviewer #1: My concerns have been addressed. I support its publication.

**Have all data underlying the figures and results presented in the manuscript been provided?**

Reviewer #1: None

PLOS authors have the option to publish the peer review history of their article (what does this mean?). If published, this will include your full peer review and any attached files.

Reviewer #1: No

**Data Deposition**

http://datadryad.org/submit?journalID=pgenetics&manu=PGENETICS-D-23-00364R1

**Press Queries**

---

## [Editor Report · Acceptance letter]

3 Jan 2024

PGENETICS-D-23-00364R1 

Growth-inhibiting effects of the unconventional plant APYRASE 7 of *Arabidopsis thaliana* influences the LRX/RALF/FER growth regulatory module 

Dear Dr Ringli, 

We are pleased to inform you that your manuscript entitled "Growth-inhibiting effects of the unconventional plant APYRASE 7 of *Arabidopsis thaliana* influences the LRX/RALF/FER growth regulatory module" has been formally accepted for publication in PLOS Genetics! Your manuscript is now with our production department and you will be notified of the publication date in due course.

With kind regards,

Anita Estes

PLOS Genetics

On behalf of:
